# Unsupervised Domain Adaptive Lane Detection via Contextual Contrast and Aggregation

## Abstract

This paper focuses on two crucial issues in domain-adaptive lane detection, i.e., how to effectively learn discriminative features and transfer knowledge across domains. Existing lane detection methods usually exploit a pixel-wise cross-entropy loss to train detection models. However, the loss ignores the difference in feature representation among lanes, which leads to inefficient feature learning. On the other hand, cross-domain context dependency crucial for transferring knowledge across domains remains unexplored in existing lane detection methods. This paper proposes a Domain-Adaptive lane detection via Contextual Contrast and Aggregation (DACCA), consisting of two key components, i.e., cross-domain contrastive loss and domain-level feature aggregation, to realize domain-adaptive lane detection. The former can effectively differentiate feature representations among categories by taking domain-level features as positive samples. The latter fuses the domain-level and pixel-level features to strengthen cross-domain context dependency. Extensive experiments show that DACCA significantly improves the detection model's performance and outperforms existing unsupervised domain adaptive lane detection methods on six datasets, especially achieving the best accuracy of 92.24% when using RTFormer on TuLane.

## 1 Introduction

Lane detection is crucial in autonomous driving and advanced driver assistance systems. Benefitting from developing convolutional neural networks, deep learning-based lane detection methods (Pan et al., 2018; Xu et al., 2020) demonstrate greater robustness and higher accuracy than traditional methods (Liu et al., 2010). To train a robust lane detection model, a high-quality dataset is necessary. However, acquiring high-quality labeled data is laborious and costly. Simulation is a low-cost way to obtain training pictures. Nevertheless, the detection performance may be degraded after transitioning from the virtual (source domain) to the real (target domain). Unsupervised domain adaptation (UDA) has been proposed to solve this problem (Saito et al., 2018; Vu et al., 2019).

Recently, UDA has been successfully applied in the image segmentation task (Vu et al., 2019; Tarvainen & Valpola, 2017), significantly improving the segmentation performance. However, applying existing unsupervised domain-adaptive segmentation methods to lane detection does not yield satisfactory results, even inferior to those of supervised training, as revealed in (Li et al., 2022). We consider the cross-entropy loss adopted in these methods only focuses on pulling similar features closer but ignores different features across categories, making these methods inefficient in learning discriminative features of different categories (Vayyat et al., 2022). Contrastive learning (He et al., 2020; Chen et al., 2020) is expected to solve this problem by appropriately selecting positive and negative samples. However, segmentation models may generate false pseudo-labels on the input image for the unlabeled target domain, causing false assignments of positive samples. On the other hand, cross-domain context dependency is essential for adaptive learning of cross-domain context information (Yang et al., 2021), which is overlooked by many existing domain adaptive lane detection methods, e.g. (Garnett et al., 2020) and (Gebele et al., 2022). In MLDA (Li et al., 2022), an Adaptive Inter-domain Embedding Module (AIEM) is proposed to aggregate contextual information, but it is limited to performing on a single image and disregards useful contextual information

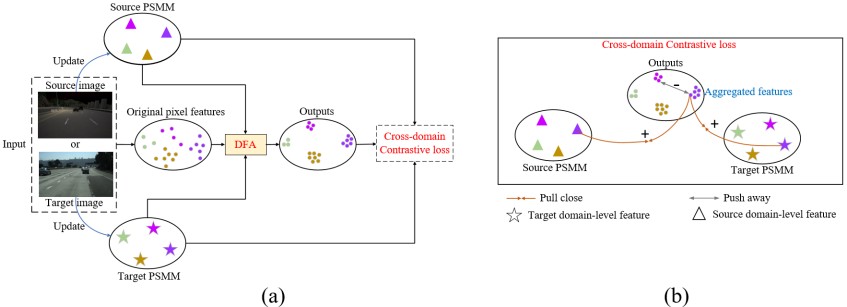

Figure 1: Diagrams of our DACCA. (a) The main flowchart, and (b) the proposed cross-domain contrastive loss, where DFA is the abbreviation of domain-level feature aggregation. Different colors in the original pixel features represent different lane features.

from other images. How to effectively leverage the potential of cross-domain context dependency in domain-adaptive lane detection remains a challenging topic.

This paper presents a novel Domain-Adaptive lane detection via Contextual Contrast and Aggregation (DACCA) to address the aforementioned issues. As shown in Figure 1, two positive sample memory modules (PSMMs) are adopted to save domain-level features for each lane in both source and target domains. We select two corresponding domain-level features as positive samples from both source and target PSMMs for each lane pixel in an input image. Subsequently, the selected domain-level features are aggregated with the original pixel feature to enrich the cross-domain contextual information. In addition, we pair the aggregated features with the source and target positive samples to avoid the false assignment of positive samples in the cross-domain contrastive loss.

The main contributions of this paper are as follows. **(1)** We propose a novel cross-domain contrastive loss to learn discriminative features and a novel sampling strategy to fully utilize the potential of contrastive loss without modifying an existing contrastive loss. **(2)** A novel domain-level feature aggregation module combining pixel-level and domain-level features is presented to enhance cross-domain context dependency, Aggregating domain-level features, instead of feature aggregation of mini-batches or individual images, is a fresh perspective. **(3)** Extensive experiments show that our method can significantly improve the baseline performance on six public datasets. Remarkably, we achieve the best results on TuLane using RTFormer (Wang et al., 2022).

## 2 RELATED WORK

**Lane detection**. Traditional lane detection mainly depends on image processing operators, e.g., Hough transforms (Liu et al., 2010). Although they can quickly achieve high detection accuracy in specific scenarios, their generalization ability is too poor to apply to complex scenarios. Deep learning-based lane detection has received increasing attention, including segmentation-based methods (Pan et al., 2018; Zheng et al., 2021) and anchor-based methods (Torres et al., 2020; Liu et al., 2021). SCNN (Pan et al., 2018) is one of the typical segmentation-based methods using a message-passing module to enhance visual evidence. Unlike pixel-wise prediction in segmentation-based methods, anchor-based methods regress accurate lanes by refining predefined lane anchors. For example, using a lightweight backbone, UFLD (Qin et al., 2020) pioneers row anchors in real-time lane detection. In this paper, we consider segmentation-based domain-adaptive lane detection.

**Unsupervised domain adaptation**. Domain adaptation has been widely studied to address the domain discrepancy in feature distribution, usually, implemented through adversarial training and self-training. Adversarial training (Gong et al., 2019) eliminates the differences in feature distribution between the source and target domains by adversarial approaches. Different from adversarial training, self-training (Sajjadi et al., 2016; Tarvainen & Valpola, 2017) trains a model in the target domain using generated pseudo labels. On the other hand, the contrastive loss is introduced as an auxiliary loss to improve the model's robustness. CDCL (Wang et al., 2023) takes labels and pseudo-labels as positive samples in the source and target domain, respectively. However, the model may generate false pseudo labels in the unlabeled target domain, leading to false positive sample assignments. There exists some works (Li et al., 2023; Wang et al., 2021; Jiang et al., 2022; Zhang et al., 2022; Melas-Kyriazi & Manrai, 2021) taking positive samples from the prototypes to achieve accu-

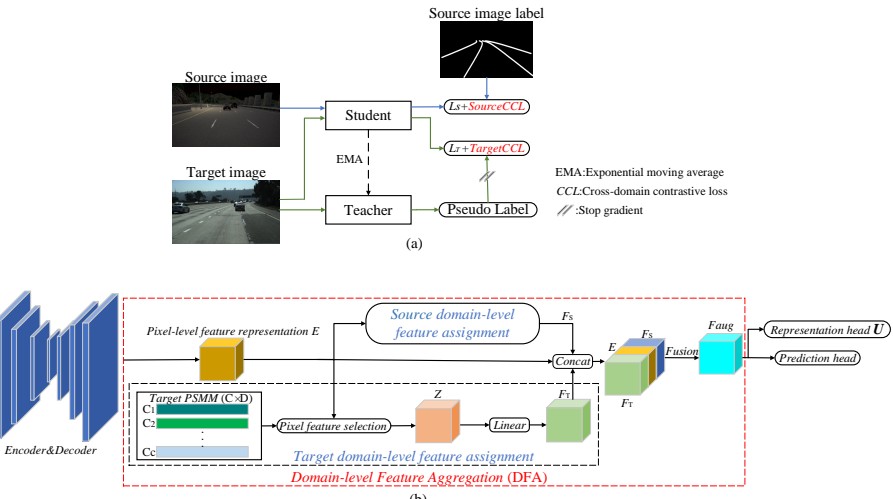

Figure 2: An overview of DACCA's framework. (a) Training pipeline of DACCA. (b) Student/Teacher model structure. The source domain-level feature assignment shares the same structure with the target domain-level feature assignment, except that a PSMM saves features from the source domain. The representation head $U$ is used to obtain the pixel-wise feature representation.

rate positive sample assignments. CONFETI (Li et al., 2023) adopts the pixel-to-prototype contrast to enhance the feature-level alignment. CONFETI only uses a prototype to save source and target domain features, but we think this way is inappropriate because the feature distribution between the two domains is different. In our work, we use two PSMMs to save features of two domains separately and take the domain-level features as positive samples. In addition, we also optimize the sample selection policy in the contrastive loss but most works ignore it.

**Unsupervised domain adaptive lane detection**. Due to the lack of a domain adaptive lane detection dataset, early studies (Garnett et al., 2020; Hu et al., 2022) focus on synthetic-to-real or simulation-to-real domain adaptation. Their generalizability in real-world scenarios is not satisfactory with low-quality synthetic and simulation images. (Gebele et al., 2022) establishes a specific dataset for domain adaptive lane detection and directly apply a general domain adaption segmentation method to this dataset. However, it does not yield good results, since conventional domain adaptive segmentation methods generally assume the presence of salient foreground objects in the image, occupying a significant proportion of the pixels. On the other hand, lane lines, which occupy a relatively small proportion of the image, do not exhibit such characteristics. To solve this problem, MLDA (Li et al., 2022) introduces an AIEM to enhance the feature representation of lane pixel by aggregating contextual information in a single image. Unfortunately, in this way, useful contextual information from other images may be ignored. Instead, we propose to aggregate the domain-level features with pixel-level features.

**Context aggregation**. Performing contextual information aggregation for pixel-level features can effectively improve segmentation performance in semantic segmentation. In supervised methods, common context information aggregation modules, e.g., ASPP (Chen et al., 2017), PSPNet (Zhao et al., 2017), OCRNet (Yuan et al., 2020), and MCIBI (Jin et al., 2021), only aggregate features within a single domain instead of both target and source domains. In UDA, some methods try to design modules to aggregate contextual features by attention mechanisms, such as cross-domain self-attention (Chung et al., 2023), and context-aware mixup (Zhou et al., 2022). However, all existing cross-domain feature aggregation methods only fuse a mini-batch of contextual features. In contrast to previous works, our method tries to simultaneously fuse features from the whole target and source domains to enhance the cross-domain context dependency.

## 3 METHOD

As illustrated in Figure 2, the network is self-trained in our DACCA, where the student model is trained in both the labeled source domain and the unlabeled target domain with pseudo-labels generated by the teacher model. DACCA has two key components, i.e., cross-domain contrastive loss and domain-level feature aggregation.

### 3.1 SELF-TRAINING

In UDA, a segmentation-based lane detection model $s_\theta$ is trained using source images $X^s = \{x_S^k\}_{k=1}^{N_s}$ with labels $Y^s = \{y_S^k\}_{k=1}^{N_s}$, to achieve a good performance on the unlabeled target images $X^t = \{x_T^k\}_{k=1}^{N_t}$, where $N_s$ and $N_t$ are the number of source and target images, respectively. $y_S^k$ is a one-hot label. Pixel-wise cross-entropy loss $L_S^k$ is adopted to train $s_\theta$ in the source domain.

$$L_S^k = -\sum_{i=1}^{H}\sum_{j=1}^{W}\sum_{c=1}^{C+1}(y_S^k)_{(i,j,c)} \times log(s_\theta(x_S^k)_{(i,j,c)}), \tag{1}$$

where $C$ is the number of lanes and class $C + 1$ denotes the background category. $H$ and $W$ are the height and width of $x_S^k$. However, when transferred to the target domain, $s_\theta$ trained in the source domain suffers from performance degradation due to the domain shift. In this paper, we adopt a self-training method (Tarvainen & Valpola, 2017) to address this issue.

As shown in Figure 2 (a), in the self-training process, we train two models, i.e., student model $s_\theta$ and teacher model $t_\theta$ to better transfer the knowledge from the source domain to the target domain. Specifically, $t_\theta$ generates the one-hot pseudo-label $y_T^k$ on the unlabeled target image $x_T^k$.

$$(y_T^k)_{(i,j,c)} = \left[c = \underset{c' \in c*}{\operatorname{argmax}}(t_\theta\left(x_T^k\right)_{(i,j,c')})\right], i \in [0, H], j \in [0, W], \tag{2}$$

where $[\cdot]$ denotes the Iverson bracket and $c*$ represents the set of all categories. To ensure the quality of pseudo-labels, we filter low-quality pseudo-labels by setting the confidence threshold $\alpha_c$, i.e.,

$$(y_T^k)_{(i,j,c)} = \begin{cases} (y_T^k)_{(i,j,c)}, & if \ (t_\theta\left(x_T^k\right)_{(i,j,c)}) \geq \alpha_c \\ 0, & otherwise \end{cases}. \tag{3}$$

$s_\theta$ is trained on both labeled source images and unlabeled target images with pseudo-labels. The same pixel-wise cross-entropy loss $L_T^k$ is used as the loss function in the target domain.

$$L_T^k = -\sum_{i=1}^{H}\sum_{j=1}^{W}\sum_{c=1}^{C+1}(y_T^k)_{(i,j,c)} \times log(s_\theta(x_T^k)_{(i,j,c)}). \tag{4}$$

During training, no gradients are backpropagated into $t_\theta$ and the weight of $t_\theta$ is updated by $s_\theta$ through Exponentially Moving Average (EMA) at every iteration $m$, denoted by,

$$t_\theta^{m+1} = \beta \times t_\theta^m + (1 - \beta) \times s_\theta^m, \tag{5}$$

where the scale factor $\beta$ is set to 0.9 empirically. After the training, we use the student model $s_\theta$ for inference and produce the final lane detection results.

### 3.2 CROSS-DOMAIN CONTRASTIVE LOSS

Since the cross-entropy loss is ineffective in learning discriminative features of different lanes, we introduce the category-wise contrastive loss (Wang et al., 2021) to solve this problem. The formulation of category-wise contrastive loss $L_{CL}$ is written as,

$$L_{CL} = -\frac{1}{C \times M}\sum_{c=1}^{C}\sum_{p=1}^{M}\log\left[\frac{e^{<V_{cp},V_c^+>/\tau}}{e^{<V_{cp},V_c^+>/\tau} + \sum_{q=1}^{N}e^{<V_{cp},V_{cpq}^->/\tau}}\right], \tag{6}$$

where $M$ and $N$ represent the numbers of anchors and negative samples, respectively. $V_{cp}$ is the feature representation of the $p$-th anchors of class $c$, used as a candidate for comparison. $V_c^+$ is the feature representation of the positive sample of class $c$. $V_{cpq}^-$ denotes the feature representation of the $q$-th negative samples of the $p$-th anchors of class $c$. $\tau$ is the temperature hyper-parameter and $\langle \cdot, \cdot \rangle$ is the cosine similarity between features from two different samples.

In the target domain, existing methods either focus on improving the form of contrastive loss (Wang et al., 2023), introducing extra hyper-parameters, or only select $V_c^+$ from the current input images (Wang et al., 2021). However, the false pseudo-labels generated by $t_\theta$ cause the incorrect positive samples assignment, making the contrastive loss ineffective in learning discriminate features of different categories. We develop a sample selection policy without modifying the existing contrastive loss to overcome the difficulty.

**Anchor Selection**. We choose anchors for each lane from a mini-batch of samples. The anchors of the $c$-th lane, $A_c$ can be selected according to,

$$A_c = \{(i,j)|GT_{(i,j)} = c, s_\theta\left(x^{in}\right)_{(i,j,c)} \geq \mu_c, i \in [0,H], j \in [0,W]\}, \tag{7}$$

$$V_c = \{V_{(i,j)}|(i,j) \in A_c\}, \tag{8}$$

where $GT$ denotes the labels in the source domain or pseudo-labels in the target domain, $x^{in}$ represents an input image, and $\mu_c$ is the threshold. We set pixels whose GT are category $c$ and whose predicted confidence are greater than $\mu_c$ as anchors to reduce the effect of hard anchors. $V \in R^{H \times W \times D}$ is the pixel-wise representation and $D$ is the feature dimension. As illustrated in Figure 2 (b), we achieve $V$ by exploiting an extra representation head $U$. $U$ shares the input with the prediction head and is only used in the training process. $V_c$ is the set of feature representation of anchors and $V_{cp} \in R^D$ is randomly selected from $V_c$.

**Positive sample selection**. To ensure the appropriate assignment of positive samples, we establish a positive sample memory module (PSMM) for each lane in both the source and target domains to save its domain-level feature, denoted as $B_{so} \in R^{C \times D}$ and $B_{ta} \in R^{C \times D}$. We initialize and update the domain-level features saved in PSMM, following MCIBI (Jin et al., 2021). This process can be found in Appendix A.2. For the $c$-th lane, we take its domain-level feature as the feature representation of the positive sample.

$$V_c^+ = B_o(c), \tag{9}$$

where $o$ is the source domain ($so$) or the target domain ($ta$).

**Negative sample selection**. We directly use pixels of a lane not labeled $c$ as the negative samples in the source domain. On the other hand, in the target domain, pixels with the lowest predicted conference for category $c$ are selected as negative samples.

$$neg\_loc_c = \left\{(i,j) \mid \underset{c' \in c*}{\arg\min}\left(s_\theta\left(x_T^k\right)_{(i,j,c')}\right) = c, i \in [0,W], j \in [0,H]\right\}, \tag{10}$$

$$neg_c = \{V_{(i,j)}|(i,j) \in neg\_loc_c\}, \tag{11}$$

where $neg\_loc_c$ and $neg_c$ denote the location and the set of feature representation of negative samples of class $c$, respectively. $V_{cpq}^- \in R^D$ is also randomly selected from $neg_c$. To compare intra-domain and inter-domain features at the same time, we propose a Cross-domain Contrastive Loss (CCL), consisting of an intra-domain contrastive learning loss $L_{inter}$ and an inter-domain contrastive learning loss $L_{intra}$.

$$CCL = L_{inter} + L_{intra}, \tag{12}$$

where $L_{inter}$ and $L_{intra}$ are the same as Eq. 6. CCL is applied in both source and target domains. For the source cross-domain contrastive loss (SCCL), the positive samples in $L_{inter}$ are the domain-level features saved in $B_{ta}$, and the positive samples in $L_{intra}$ are the domain-level features saved in $B_{so}$. The positive samples in the target cross-domain contrastive loss (TCCL) are opposite to SCCL. The overall loss of DACCA is,

$$Loss = \frac{1}{N_s}\sum_{k=1}^{N_s}(\lambda_c \times SCCL^k + L_S^k) + \frac{1}{N_t}\sum_{k=1}^{N_t}(\lambda_c \times TCCL^k + L_T^k), \tag{13}$$

where $\lambda_c$ is the scale factor, which is set to 0.1 empirically.

## 3.3 DOMAIN-LEVEL FEATURE AGGREGATION

Cross-domain context dependency is essential to transfer knowledge across domains. Cross-domain Contextual Feature Aggregation (CCFA) is an effective way to achieve cross-domain context dependency. Existing CCFA methods (Yang et al., 2021; Zhou et al., 2022; Chung et al., 2023) only aggregate a mini-batch of features. We argue that aggregating features from a whole domain is more beneficial. As shown in Figure 2 (b), Domain-level Feature Aggregation (DFA) aims to fuse the domain-level features into the pixel-level representation. DFA contains two key components, i.e., source and target domain-level feature assignment. The process is the same for both. We take the target domain-level feature assignment as an example to depict the process.

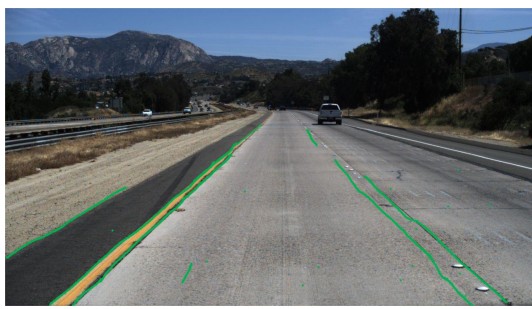

Figure 3: Location of unreliable background pixels in green.

**Pixel feature selection**. To select the corresponding domain-level feature for each lane pixel, we propose the pixel feature selection. We first obtain the predicted category at location $(i, j)$ by,

$$P = \underset{c' \in c*}{\operatorname{argmax}}(\operatorname{Softmax}(\operatorname{Conv}(E))_{(i,j,c')}), i \in [0, W], j \in [0, H], \tag{14}$$

where $E \in R^{H \times W \times D}$ represents the feature map, containing the pixel-level feature representation. 1×1 convolution (termed as $\operatorname{Conv}$) is adopted to change the channels of $E$ to $C + 1$. $P \in R^{H \times W}$ saves the predicted category at each location of $E$. Then, we build a feature map $Z$ whose pixel values are zero and whose size and dimension are the same as $E$. We assign the pixel-wise feature to $Z$ using the domain-level feature.

$$Z_{(i,j)} = B_{ta}\left(P_{(i,j)}\right), P_{(i,j)} \neq C + 1, i \in [0, W], j \in [0, H]. \tag{15}$$

After the assignment, $Z$ is a domain-level feature map. Here, the lane pixels on $E$ predicted as the background in training are called unreliable background pixels (UBP). For example, as illustrated in Figure 3, UBP is mainly located at the edge of the lane. However, the features of UBP can not be augmented since domain-level features are only aggregated for the foreground pixels. To refine the features of UBP, we also perform further feature aggregation on UBP.

Specifically, the predicted confidence of the UBP is usually low, hence we distinguish UBP from reliable background pixels by setting confidence threshold $\varepsilon$. The UBP is defined as,

$$UBP = \{(i,j)|pred_{(i,j)} < \varepsilon, P_{(i,j)} = C + 1, i \in [0, W], j \in [0, H]\}, \tag{16}$$

where $pred_{(i,j)}$ is the confidence of the predicted category at location $(i, j)$. $pred_{(i,j)}$ is obtained by: $pred_{(i,j)} = \underset{c' \in c*}{\max}\left(\operatorname{Softmax}(\operatorname{Conv}(E))_{(i,j,c')}\right)$. We choose the category with the lowest Euclidean distance as the pseudo category of UBP and use domain-level feature of pseudo category to instantiate UBP in $Z$.

$$P_{(i,j)} = \underset{c' \in c*}{\operatorname{argmin}}\left(dis\left(E_{(i,j)}^{UBP}, B_{ta}\left(c'\right)\right)\right), (i,j) \in UBP, \tag{17}$$

$$Z_{(i,j)} = B_{ta}(P_{(i,j)}), (i,j) \in UBP, \tag{18}$$

where $E_{(i,j)}^{UBP}$ is the feature representation of UBP at location $(i, j)$ in $E$, and $dis$ is used to calculate the Euclidean distance between the feature representation of UBP and the domain-level feature.

Thereafter, we adopt a linear layer to extract features along the channel dimension in $Z$ to obtain the output of target domain-level feature assignment $F_T$. In the same process, we replace the target PSMM with the source PSMM to obtain the feature $F_S$. $F_S$, $F_T$, and $E$ are concatenated along the channel dimension and fused by a 1×1 convolution to enrich the cross-domain context information of $E$.

$$F_{aug} = \operatorname{Conv}(\varphi(E, F_S, F_T)), \tag{19}$$

where $F_{aug} \in R^{H \times W \times D}$ is the aggregated features and $\varphi$ is the concatenate operation.

## 4 EXPERIMENTS

### 4.1 EXPERIMENTAL SETTING

We provide the experimental setting including datasets and implementation details in Appendix A.1.

Table 1: Results of critical components.

| Source-only | SCCL | Self-Training | TCCL | DFA | UBP | Accuracy(%) | FP(%) | FN(%) |
|---|---|---|---|---|---|---|---|---|
| ✓ | | | | | | 77.42 | 58.29 | 54.19 |
| ✓ | ✓ | | | | | 79.63 | 53.41 | 50.00 |
| ✓ | ✓ | ✓ | | | | 80.76 | 49.39 | 47.50 |
| ✓ | ✓ | ✓ | ✓ | | | 81.77 | 48.36 | 45.06 |
| ✓ | ✓ | ✓ | ✓ | ✓ | | 82.43 | 44.53 | 42.89 |
| ✓ | ✓ | ✓ | ✓ | ✓ | ✓ | 83.99 | 42.27 | 40.10 |

## 4.2 ABLATION STUDY

We ablate the key components of DACCA and use SCNN with ResNet50 (He et al., 2016) as the detection model. If not specified, all ablation studies are conducted on TuLane. Additional ablation study can be found in Appendix A.3.

**Effectiveness of cross-domain contrastive learning (CCL)**. In Table 1, when only source domain data are used in supervised learning, SCCL prompts the accuracy from 77.42% to 79.63%. It also indicates that our SCCL works for supervised training. On the other hand, the accuracy increases by 1.01%, i.e., from 80.76% to 81.77%, if TCCL is adopted. T-SNE visualization in Figure A4 (c) of Appendix A.4 shows that the model with CCL can learn more discriminative features.

**Effectiveness of domain-level feature aggregation (DFA)**. In Table 1, DFA can improve the detection accuracy from 81.77% to 82.43%. As for feature aggregation of UBP, the accuracy is further increased by 1.56% (83.99% vs. 82.43%). Also, we can observe a significant adaptation of the source and target domain features in Figure A2 (c) of Appendix A.4, which validates the effectiveness of domain-level feature aggregation.

Table 2: Generalizability of different methods. The symbol * indicates source domain only.

| Model | Backbone | Accuracy/% | FP/% | FN/% |
|---|---|---|---|---|
| SCNN* | ResNet50 | 77.42 | 58.29 | 54.19 |
| SCNN+DACCA | ResNet50 | **83.99** | **42.27** | **40.10** |
| ERFNet (Romera et al., 2017)* | ERFNet | 83.30 | 37.46 | 37.55 |
| ERFNet+DACCA | ERFNet | **90.47** | **30.66** | **18.16** |
| RTFormer (Wang et al., 2022)* | RTFormer-Base | 87.24 | 26.78 | 25.17 |
| RTFormer+DACCA | RTFormer-Base | **92.24** | **15.10** | **12.58** |

**Generalizability of different methods**. As shown in Table 2, our method can be integrated into various segmentation-based lane detection methods. In SCNN, using our method can increase the accuracy by 6.57% and decrease FP and FN by 16.02% and 14.09%, respectively. Also, in the lightweight model ERFNet, the accuracy rises by 7.17%, and FP and FN drop by 6.8% and 19.39%. Finally, in the Transformer-based method RTFormer, our method significantly improves the detection performance, in terms of accuracy, FP, and FN.

**Comparison with existing contrastive loss variants**. In Figure 4 (a), CCL is evaluated against other contrastive loss variants in UDA. In turn, we replace CCL in DACCA with CDCL, ProCA (Jiang et al., 2022), CONFETI (Li et al., 2023), and SePiCo (Xie et al., 2023). Compared with ProCA and CONFETI, CCL increases the accuracy by 2.58% (81.77% vs. 79.19%) and 1.9% (81.77% vs. 79.87%), respectively. The reason may be that both ProCA and CONFETI ignore the differences in feature distribution between the source domain and target domain and only use a prototype to represent the features of the two domains. Moreover, CCL overwhelms SePiCo regarding accuracy. It attributes to SePiCo only taking domain-level features from the source domain as the positive samples but ignoring the samples from the target domain.

**Comparison with existing cross-domain context aggregation**. We substitute the DFA with Cross-domain (Yang et al., 2021) and Self-attention module (SAM) (Chung et al., 2023)—the latter aggregate features in a mini-batch. The superiority of the DFA is shown in Figure 4 (b). DFA performs better than Cross-domain and SAM, e.g., prompts the accuracy by 0.46% (83.51% vs. 83.05%) and 0.72% (83.51% vs. 82.79%), respectively. From the T-SNE visualization in Figure A3 of Appendix A.4, we can see that DFA aligns the features of two domains better. The results demonstrate that aggregating features from the whole domain is more effective than from a mini-batch.

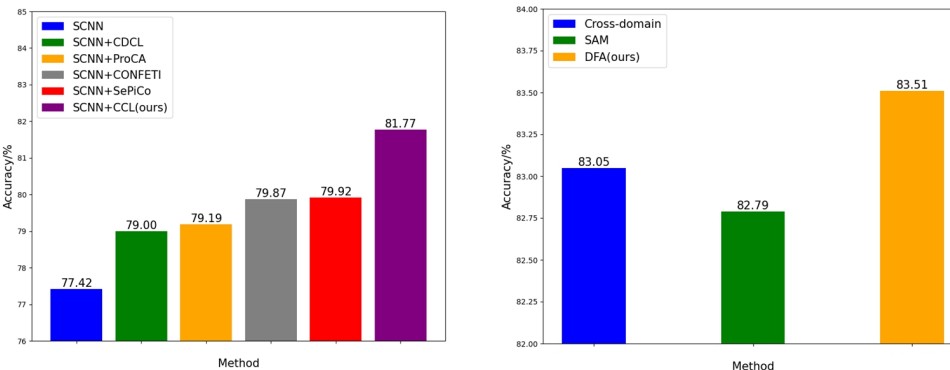

Figure 4: Accuracy comparison with counterparts of key peer components. (a) Comparison among existing contrastive loss variants. (b) Comparison among existing cross-domain context aggregation.

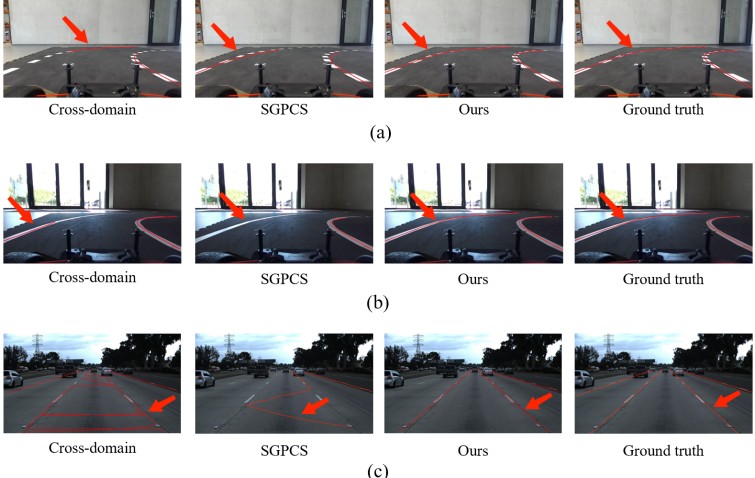

Figure 5: Visualization result comparison among cross-domain, SGPCS, and our method. Results on (a) MuLane, (b) MoLane, and (c) TuLane.

Table 3: Performance comparison on TuLane.

| Method | Detection model | Backbone | Accuracy/% | FP/% | FN/% |
|---|---|---|---|---|---|
| DANN (Gebele et al., 2022) | ERFNet | ERFNet | 86.69 | 33.78 | 23.64 |
| ADDA (Gebele et al., 2022) | ERFNet | ERFNet | 87.90 | 32.68 | 22.33 |
| SGADA (Gebele et al., 2022) | ERFNet | ERFNet | 89.09 | 31.49 | 21.36 |
| SGPCS (Gebele et al., 2022) | ERFNet | ERFNet | 89.28 | 31.47 | 21.48 |
| SGPCS | RTFormer | RTFormer-Base | 90.78 | 28.44 | 15.66 |
| SGPCS | UFLD | ResNet18 | 91.55 | 28.52 | 16.16 |
| LD-BN-ADAPT (Bhardwaj et al., 2023) | UFLD | ResNet18 | 92.00 | - | - |
| MLDA (Li et al., 2022) | ERFNet | ERFNet | 88.43 | 31.69 | 21.33 |
| MLDA+CCL | ERFNet | ERFNet | 89.00 | 30.53 | 20.42 |
| MLDA+DFA | ERFNet | ERFNet | 89.45 | 30.22 | 20.02 |
| PyCDA (Lian et al., 2019) | ERFNet | ERFNet | 86.73 | 31.26 | 24.13 |
| Cross-domain (Yang et al., 2021) | ERFNet | ERFNet | 88.21 | 29.17 | 22.27 |
| Maximum Squares (Chen et al., 2019) | ERFNet | ERFNet | 85.98 | 30.20 | 26.85 |
| DACCA | ERFNet | ERFNet | 90.47 | 30.66 | 18.16 |
| DACCA | RTFormer | RTFormer-Base | **92.24** | **15.10** | **12.58** |

## 4.3 COMPARISON WITH STATE-OF-THE-ART METHODS

**Performance on TuLane**. The results on TuLane are shown in Table 3. When ERFNet is used as the detection model, our method performs better than other methods. For instance, our method

Table 4: Performance comparison on "OpenLane" to "CULane".

| Method | Normal | Crowded | Night | No line | Shadow | Arrow | Dazzle | Curve | Cross | Total |
|---|---|---|---|---|---|---|---|---|---|---|
| Advent (Li et al., 2022) | 51.2 | 24.5 | 21.5 | 19.9 | 16.9 | 34.7 | 27.2 | 35.3 | 5789 | 31.7 |
| PyCDA (Lian et al., 2019) | 42.4 | 20.6 | 14.7 | 15.9 | 14.4 | 28.6 | 19.5 | 30.8 | **4452** | 26.3 |
| Maximum Squares (Chen et al., 2019) | 51.4 | 28.4 | 22.1 | 19.7 | 20.9 | 40.8 | 28.1 | 39.3 | 9813 | 31.8 |
| MLDA (Li et al., 2022) | 62.0 | 38.0 | 28.5 | 21.9 | 24.1 | 50.3 | 31.7 | **44.5** | 11399 | 38.8 |
| DACCA | **64.9** | **39.6** | **29.3** | **25.1** | **26.3** | **52.8** | **34.1** | 43.5 | 7158 | **43.0** |

Table 5: Performance comparison on "CULane" to "Tusimple".

| Method | Detection model | Backbone | Accuracy/% | FP/% | FN/% |
|---|---|---|---|---|---|
| Advent (Li et al., 2022) | ERFNet | ERFNet | 77.1 | 39.7 | 43.9 |
| PyCDA (Lian et al., 2019) | ERFNet | ERFNet | 80.9 | 51.9 | 45.1 |
| Maximum Squares (Chen et al., 2019) | ERFNet | ERFNet | 76.0 | 38.2 | 42.8 |
| MLDA (Li et al., 2022) | ERFNet | ERFNet | 89.7 | 29.5 | 18.4 |
| DACCA | ERFNet | ERFNet | **92.1** | **26.7** | **14.6** |

outperforms MLDA in terms of accuracy by 2.04% (90.47% vs. 88.43%). Besides, using our CCL and DFA, the performance of MLDA gains consistent improvement. It indicates our sample selection policy is more effective than designing complicated loss functions, and DFA has a stronger domain adaptive ability than AIEM in MLDA. Regarding FN metrics, our method is 5.97% and 4.11% lower than PyCDA and Cross-domain, respectively. Significantly, when using the Transformer model RTFormer, DACCA outperforms the state-of-the-art SGPCS (92.24% vs. 91.55%) and achieves the best experimental results on TuLane in similar settings.

**Performance on OpenLane to CULane**. To further validate our method's generalization ability, we carry out experiments transferring from OpenLane to CULane to demonstrate a domain adaptation between difficult real scenarios. As shown in Table 4, our method delivers 4.2% enhancement (43.0% vs. 38.8%) compared to the state-of-the-art MLDA. Our DACCA surpasses the existing methods in most indicators and also all these results reflect its outperformance.

**Performance on CULane to Tusimple**. As presented in Table 5, our DACCA achieves the best performance on "CULane to Tusimple". For instance, DACCA increases the accuracy from 89.7% to 92.1% compared with the state-of-the-art methbod MLDA. It indicates our DACCA can perform well on the domain adaptation from difficult scene to simple scene.

**Qualitative evaluation**. We display the visualization comparison results between Cross-domain, SGPCS, and our method in Figure 5. In Figure 5 (c), our method predicts more smooth lanes than the other methods in the urban scenario. Our method can detect the complete lanes in the real-world scene in Figure 5 (a) and 5 (b). Qualitative results demonstrate that our method can effectively transfer knowledge across different domains.

## 5 CONCLUSION

This paper presents a novel unsupervised domain-adaptive lane detection via contextual contrast and aggregation (DACCA), in which learning discriminative features and transferring knowledge across domains are exploited. Firstly, we create the positive sample memory module to preserve the domain-level features of the lane. Then, we propose a cross-domain contrastive loss to improve feature discrimination of different lanes by a novel sample selection strategy without modifying the form of contrastive loss. Finally, we propose the domain-level feature aggregation to fuse the domain-level features with the pixel-level features to enhance cross-domain context dependency. Experimental results show that our approach achieves the best performance on the TuLane dataset. On the MuLane and MoLane datasets, our method outperforms existing unsupervised domain-adaptive segmentation-based lane detection methods.

Although DACCA is implemented upon the segmentation-based lane detection, it holds potential for application in other lane detection methods, e.g., keypoint-based and transformer-based approaches. Our future work is to explore this aspect.

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

## A  APPENDIX

### A.1  EXPERIMENTAL SETTINGS

**Datasets**. We conduct extensive experiments to examine DACCA on six datasets for lane detection tasks, i.e., TuLane (Stuhr et al., 2022), MoLane (Stuhr et al., 2022), MuLane (Stuhr et al., 2022), CULane (Pan et al., 2018), Tusimple (tus), and OpenLane (Chen et al., 2022). The source domain of the TuLane dataset uses 24,000 labeled simulated images as the training set, and the target domain images derives from the Tusimple dataset. The source domain of the MoLane dataset uses 80,000 labeled simulated images as the training set, and the target domain training set is adopted from the real scenes and contains 43,843 unlabeled images. The MuLane dataset mixes the TuLane and MuLane datasets are uniformly blended. The source domain of MuLane dataset uses 48000 labeled simulated images as the training set, and the target domain combines the Tusimple and MoLane target domains.

Following (Li et al., 2022), we conduct the experiments on "CULane to Tusimple" and "Tusumple and CULane". "TuSimple to CULane" means that the source domain is TuSimple and the target domain is CULane. To further validate the effectiveness of our method on the domain adaptation cross difficult scenes, we carry out the experiments on "CULane to OpenLane" and "OpenLane to CULane".

**Evaluation metrics**. For TuLane, MuLane. MoLane, and Tusimple datasets. We use three official indicators to evaluate the model performance for three datasets: Accuracy, false positives (FP), and false negatives (FN). Accuracy is defined by $Accuracy = \frac{p_c}{p_y}$, where $p_c$ denotes the number of correct predicted lane points and $p_y$ is the number of ground truth lane points. A lane point is regarded as correct if its distance is smaller than the given threshold $t_{pc} = \frac{20}{cos(a_{yl})}$, where $a_{yl}$ represents the angle of the corresponding ground truth lane. We measure the rate of false positives with $FP = \frac{l_f}{l_p}$ and the rate of false positives with $FN = \frac{l_m}{l_y}$, where $l_f$ is the number of mispredicted lanes, $l_p$ is the number of predicted lanes, $l_m$ is the number of missing lanes and $l_y$ is the number of ground truth lanes. Following (Stuhr et al., 2022), we consider lanes as mispredicted if the Accuracy $< 85\%$. For CULane and OpenLane, we adopt the F1 score to measure the performance, $F_1 = \frac{2 \times Precision \times Recall}{Precision + Recall}$, where $Precision = \frac{TP}{TP+FP}$ and $Recall = \frac{TP}{TP+FN}$, where $TP$ denote the true positives.

**Implementation details**. We update the learning rate by the Poly policy with power factor $1 - (\frac{iter}{total_i iter})^{0.9}$. We select the AdamW optimizer with the initial learning rate 0.0001. We adopt the data augmentation of random rotation and flip for TuLane, and random horizontal flips and random affine transforms (translation, rotation, and scaling) for MuLane and MoLane. The training epochs on the TuLane, MuLane, and MoLane are 30, 20, and 20 respectively. We set the threshold for filtering false pseudo labels $\alpha_c$ to 0.3 during domain adaptation. The threshold for selecting anchors $\mu_c$ and UBP $\varepsilon$ are 0.2 and 0.7, respectively. The number of anchors $M$ and negative samples $N$ in the cross-domain contrastive loss are 256 and 50, respectively. Temperature hyper-parameter $\tau$ is set to 0.07 empirically. The feature dimension $D$ is 128. The optimizer and update policy of the learning rate are the same as those in pretraining. All images are resized to 384×800. All experiments are conducted on a single Tesla V100 GPU with 32 GB memory. DACCA is implemented based on PPLanedet (Zhou, 2022).

### A.2  POSITIVE SAMPLE MEMORY MODULE

Positive sample memory module (PSMM) is introduced to store domain-level features for each lane in both source and target domains. The process of initializing and updating features is the same for source and target PSMM. We take the target PSMM as an example to describe this process.

**Feature initialization**. MCIBI (Jin et al., 2021) selects the feature representation of one pixel for each lane to initialize the feature in PSMM. However, this way may bring out false feature initialization due to false pseudo labels. For the $c$-th lane, we initialize its feature in PSMM using the center of the features of all anchors, expressed by,

$$B_{ta}(c) = \frac{1}{|V_c|} \sum_{n_c \in V_c} n_c, \tag{20}$$

where $|V_c|$ denotes the number of anchors and $n_c$ is the feature representation of anchors in $V_c$.

**Feature update**. The features in target PSMM are updated through the EMA after each training iteration $m$,

$$(B_{ta}(c))_m = t_{m-1} \times (B_{ta}(c))_{m-1} + (1 - t_{m-1}) \times \partial((V_c)_{m-1}), \quad (21)$$

where $t$ is the scale factor and $\partial$ is used to transform $V_c$ to obtain the feature with the same size as $(B_{ta}(c))_{m-1}$. Following MCIBI, we adopt the polynomial annealing policy to schedule $t$,

$$t_m = (1 - \frac{m}{T})^p \times (t_0 - \frac{t_0}{100}) + \frac{t_0}{100}, m \in [0, T], \quad (22)$$

where $T$ is the total number of training iterations. We set both $p$ and $t_0$ as 0.9 empirically. To implement $\partial$, we first compute the cosine similarity vector $S_c$ between the feature representation of anchors in $(V_c)_{m-1}$ and $(B_{ta}[c])_{m-1}$, as below,

$$S_c(i) = \frac{(V_c)_{m-1}(i) \times (B_{ta}(c))_{m-1}}{\|(V_c)_{m-1}(i)\|_2 \times \|(B_{ta}(c))_{m-1}\|_2}, i \in [1, |V_c|], \quad (23)$$

where we use $(i)$ to index the element in $S_c$ or feature representation in $(V_c)_{m-1}$. Then, we obtain the output of $\partial((V_c)_{m-1})$ by,

$$\partial((V_c)_{m-1}) = \sum_{i=1}^{|V_c|} \frac{1 - S_c(i)}{\sum_{j=1}^{|V_c|}(1 - S_c(j))} \times (V_c)_{m-1}(i). \quad (24)$$

For the source PSMM, features in $V_c$ come from the source domain.

### A.3 ADDITIONAL ABLATION STUDY

In this section, we provide additional ablation studies about hyper-parameters. If not specified, we still adopt the SCNN with the ResNet50 backbone as the detection model and experiments are conducted on TuLane.

**The number of anchors** $M$. We study the influence of the number of anchors $M$ and the results are shown in Figure A1 (a). It can be observed that the model achieves the best performance when $M$ is 256. Besides, It causes extra computational burden when $M$ increases. Considering accuracy and computational burden, we set $M$ as 256.

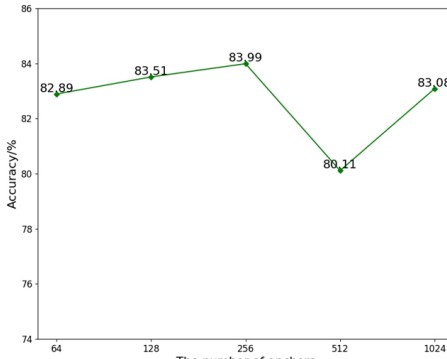
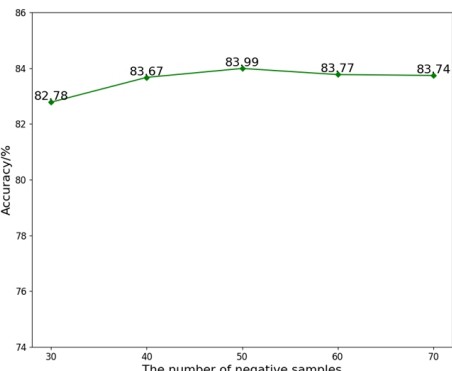

Figure A1: Hyper-parameter study. (a) Study of the number of anchors. (b) Study of the number of negative samples.

**The number of negative samples** $N$. Figure A1 (b) shows the influence of the number of negative samples. When $N$ is 50, model achieves the best performance. We can also see that as $N$ increases, the accuracy does not always improve, indicating that excessive negative samples can degrade performance.

**The threshold for selecting anchors** $\mu_c$. We study the threshold for selecting anchors $\mu_c$. As shown in Table A1, setting the anchor selection threshold can avoid hard anchors compared with anchor selection without the threshold (83.99% vs. 80.97%). However, when the threshold is too high, available anchors shrink, leading to performance degradation (83.99% vs. 80.80%). Hence, we set $\mu_c$ to 0.2.

<div>

Table A1: The threshold for selecting anchors.

| $\mu_c$ | Accuracy/% | FP/% | FN/% |
|---|---|---|---|
| 0.0 | 80.97 | 51.72 | 46.95 |
| 0.1 | 81.27 | 51.45 | 46.84 |
| 0.2 | **83.99** | **42.27** | **40.10** |
| 0.3 | 80.79 | 50.81 | 46.52 |
| 0.4 | 80.80 | 52.14 | 49.25 |

</div>

<div>

Table A2: The threshold for selecting UBP.

| $\varepsilon$ | Accuracy/% | FP/% | FN/% |
|---|---|---|---|
| - | 83.32 | 46.83 | 40.76 |
| 0.5 | 83.41 | 44.79 | 40.67 |
| 0.6 | 83.38 | 46.67 | 40.17 |
| 0.7 | **83.99** | **42.27** | **40.10** |
| 0.8 | 82.20 | 47.55 | 43.53 |
| 0.9 | 81.66 | 48.49 | 45.34 |

</div>

**The threshold for selecting UBP**. We can see that without the feature refinement of UBP, accuracy is only 83.32% in Table A2. When $\varepsilon$ is 0.7, model achieves the best performance. It has little effect on model performance when $\varepsilon$ is too low. This is attributed to the small number of UBP. When $\varepsilon$ is too high, many background pixels are wrongly regarded as UBP, causing the negative effect.

Table A3: The threshold for filtering false pseudo labels.

| $\alpha_c$ | Accuracy/% | FP/% | FN/% |
|---|---|---|---|
| 0.1 | 81.39 | 52.36 | 48.02 |
| 0.2 | 82.37 | 48.25 | 45.81 |
| 0.3 | **83.99** | **42.27** | **40.10** |
| 0.4 | 83.58 | 44.41 | 42.10 |
| 0.5 | 83.49 | 45.09 | 42.79 |

Table A4: The way of feature aggregation

| Way | Accuracy/% | FP/% | FN/% |
|---|---|---|---|
| Add | 77.92 | 52.72 | 54.04 |
| Weighted add | 80.05 | 49.15 | 48.23 |
| Concatenation | **83.99** | **42.27** | **40.10** |

**The threshold for filtering false pseudo labels**. We study the threshold for filtering false pseudo labels $\alpha_c$ and results are shown in Table A3. When $\alpha_c$ is low, false pseudo labels have a greater impact on performance. If $\alpha_c$ is too high, the number of pseudo labels is too small, providing insufficient supervision signals. Therefore, we set $\alpha_c$ to 0.3.

**The way of feature fusion**. We study the way of feature fusion in Table A4. Add denotes for element-wise adding $E$, $F_S$, and $F_T$. Compared with add, concatenation gains 6.07% accuracy improvements. The reason may be that Add directly changes the original pixel features but concatenation does not. Weighted add means adding $E$, $F_S$, and $F_T$ weightedly where weights are predicted by a $1 \times 1$ convolution. Concatenation overwhelms Weighted add regarding accuracy, FN, and FP. We adopt the concatenation as the way of feature fusion.

## A.4 VISUALIZATION OF CROSS-DOMAIN FEATURES

**T-SNE visualization of the key components**. As shown in Figure A2 (a). There is a slight adaptation of cross-domain features when model is only trained in the source domain. Learned cross-domain features are aligned better using our proposed CCL in Figure A2 (b). However, since CCL is a pixel-wise contrast, it can lead to the separation of the feature space due to lack of contextual information. To solve this problem, we enhance the links between cross-domain features by introducing domain-level feature aggregation (DFA). DFA incorporate cross-domain contextual information into the pixel-wise feature and effectively address the separation of the feature space in Figure A2 (c).

**T-SNE visualization of different cross-domain context aggregation methods**. Compared with Cross-domain (Yang et al., 2021) and Self-attention module (SAM) (Chung et al., 2023), DACCA aligns source and target domain features better in Figure A3, indicating domain-level features can provide more cross-domain knowledge than features from a mini-batch.

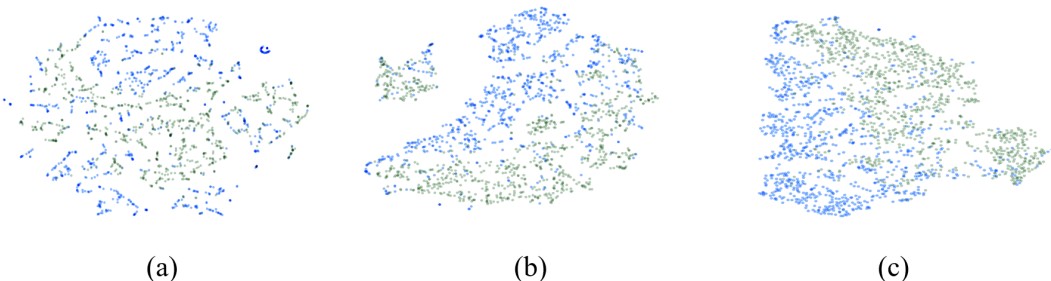

Figure A2: T-SNE visualization of the key components. (a) SCNN trained with only source domain. (b) SCNN trained with SCCL and TCCL. (c) SCNN trained with SCCL, TCCL, and DFA. Blue and green color represent the source and target domain, respectively.

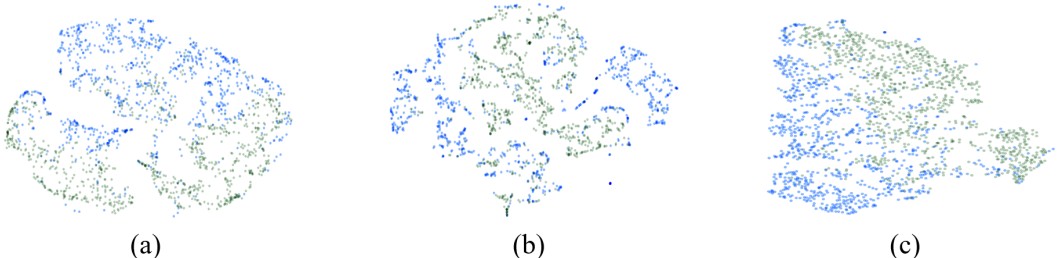

Figure A3: T-SNE visualization of different cross-domain context aggregation methods. (a) Cross-domain. (b) SAM. (c) DACCA.

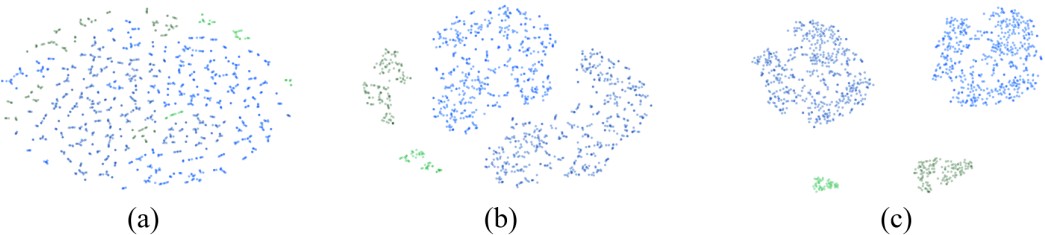

Figure A4: T-SNE visualization of different loss functions. (a) Cross-entropy loss. (b) SePiCo. (c) CCL. Different colors represent different lane features.

**T-SNE visualization of different loss functions**. As shown in Figure A4, our CCL learns more discriminative features than SePiCo, indicating that our sample selection policy is effective. Besides, cross-entropy is inefficient in discriminating features of different categories. Our CCL can effectively compensate for the deficiency of cross-entropy loss.

## A.5    ADDITIONAL EXPERIMENTS

**Tusimple to CULane**. We conduct the experiments on the domain adaptation from simple scene and difficult scene and result are shown in Table A5. DACCA demonstrates consistent performance advantages.

**CULane to OpenLane**. From Table A6, we can see that DACCA achieves the best performance and gains 5.3% F1 score improvement. The results on domain adaptation cross difficult scenes manifest the effectiveness and generalizability of our DACCA.

**Performance on MoLane**. Next, our method is tested on MoLane. By observing Table A7, we can conclude that DACCA is superior to existing unsupervised domain-adaptive lane detection methods. Specifically, DACCA improves the accuracy by 2.22% against SGPCS (93.50% vs. 91.28%). Moreover, using ERFNet as the detection model, DACCA improves the accuracy by 4.37% (90.52%

Table A5: Performance comparison on "Tusimple" to "CULane". Detection model and backbone are still ERFNet. We only report the number of false positives for Cross category following (Li et al., 2022).

| Method | Normal | Crowded | Night | No line | Shadow | Arrow | Dazzle | Curve | Cross | Total |
|---|---|---|---|---|---|---|---|---|---|---|
| Advent (Li et al., 2022) | 49.3 | 24.7 | 20.5 | 18.4 | 16.4 | 34.4 | 26.1 | 34.9 | 6257 | 30.4 |
| PyCDA (Lian et al., 2019) | 41.8 | 19.9 | 13.6 | 15.1 | 13.7 | 27.8 | 18.2 | 29.6 | **4422** | 25.1 |
| Maximum Squares (Chen et al., 2019) | 50.5 | 27.2 | 20.8 | 19.0 | 20.4 | 40.1 | 27.4 | 38.8 | 10324 | 31.0 |
| MLDA (Li et al., 2022) | 61.4 | 36.3 | 27.4 | 21.3 | 23.4 | 49.1 | 30.3 | 43.4 | 11386 | 38.4 |
| DACCA | **64.6** | **39.1** | **28.6** | **24.5** | **25.8** | **52.0** | **33.4** | 42.9 | 8517 | **41.9** |

Table A6: Performance comparison on "CULane" to "OpenLane". Detection model and backbone are still ERFNet.

| Method | All | Up&Down | Curve | Extreme Weather | Night | Intersection | Merge&Split |
|---|---|---|---|---|---|---|---|
| Advent (Li et al., 2022) | 17.3 | 12.6 | 15.8 | 20.7 | 16.9 | 9.4 | 15.6 |
| PyCDA (Lian et al., 2019) | 17.0 | 12.8 | 15.4 | 19.2 | 16.0 | 9.6 | 14.9 |
| Maximum Squares (Chen et al., 2019) | 18.9 | 13.4 | 16.0 | 21.4 | 17.3 | 11.0 | 16.8 |
| MLDA (Li et al., 2022) | 22.3 | 18.4 | 20.3 | 24.8 | 21.4 | **15.8** | 21.0 |
| DACCA | **27.6** | **25.0** | **25.3** | **31.8** | **27.0** | 14.1 | **23.9** |

Table A7: Performance comparison on MoLane. Symbol * indicates source domain only.

| Method | Detection model | Backbone | Accuracy/% | FP/% | FN/% |
|---|---|---|---|---|---|
| DANN (Gebele et al., 2022) | ERFNet | ERFNet | 85.65 | 22.25 | 22.25 |
| ADDA (Gebele et al., 2022) | ERFNet | ERFNet | 87.85 | 18.61 | 18.66 |
| SGADA (Gebele et al., 2022) | ERFNet | ERFNet | 89.46 | 15.13 | 15.13 |
| SGPCS (Gebele et al., 2022) | ERFNet | ERFNet | 90.08 | 12.16 | 12.16 |
| SGPCS (Gebele et al., 2022) | RTFormer | RTFormer-Base | 91.28 | 8.69 | 8.69 |
| LD-BN-ADAPT (Bhardwaj et al., 2023) | UFLD | ResNet18 | 92.68 | - | - |
| MLDA (Li et al., 2022) | ERFNet | ERFNet | 89.97 | 12.33 | 15.42 |
| PyCDA (Lian et al., 2019) | ERFNet | ERFNet | 87.40 | 17.59 | 18.10 |
| Cross-domain (Yang et al., 2021) | ERFNet | ERFNet | 88.57 | 15.16 | 17.41 |
| Maximum Squares (Chen et al., 2019) | ERFNet | ERFNet | 87.22 | 21.31 | 27.85 |
| DACCA* | ERFNet | ERFNet | 86.15 | 23.85 | 29.50 |
| DACCA | ERFNet | ERFNet | 90.52 | 7.00 | 13.95 |
| DACCA* | RTFormer | RTFormer-Base | 86.77 | 20.6 | 26.9 |
| DACCA | RTFormer | RTFormer-Base | **93.50** | **6.26** | **7.25** |

Table A8: Performance comparison on MuLane. Symbol * indicates source domain only.

| Method | Detection model | Backbone | Accuracy/% | FP/% | FN/% |
|---|---|---|---|---|---|
| DANN (Gebele et al., 2022) | ERFNet | ERFNet | 84.01 | 38.31 | 36.30 |
| ADDA (Gebele et al., 2022) | ERFNet | ERFNet | 85.99 | 29.38 | 28.59 |
| SGADA (Gebele et al., 2022) | ERFNet | ERFNet | 85.26 | 29.13 | 28.73 |
| SGPCS (Gebele et al., 2022) | ERFNet | ERFNet | 86.92 | 27.49 | 28.39 |
| SGPCS (Gebele et al., 2022) | RTFormer | RTFormer-Base | 88.02 | 23.98 | 25.80 |
| LD-BN-ADAPT (Bhardwaj et al., 2023) | UFLD | ResNet18 | 89.88 | - | - |
| MLDA (Li et al., 2022) | ERFNet | ERFNet | 87.28 | 32.59 | 30.06 |
| PyCDA (Lian et al., 2019) | ERFNet | ERFNet | 86.01 | 35.15 | 34.17 |
| Cross-domain (Yang et al., 2021) | ERFNet | ERFNet | 85.74 | 32.10 | 37.42 |
| Maximum Squares (Chen et al., 2019) | ERFNet | ERFNet | 84.26 | 42.59 | 49.67 |
| DACCA* | ERFNet | ERFNet | 83.28 | 47.17 | 55.21 |
| DACCA | ERFNet | ERFNet | 87.93 | 25.95 | 27.08 |
| DACCA* | RTFormer | RTFormer-Base | 84.19 | 37.88 | 39.20 |
| DACCA | RTFormer | RTFormer-Base | **90.14** | **15.11** | **17.14** |

vs. 86.15%) compared to the model using only source domain data. It is worth mentioning that if the Transformer model, RTFormer, is used as the detection model, the detection accuracy can be prompted by 6.73% (93.50% vs. 86.77%).

**Performance on MuLane**. To further validate our method's generalization ability, we carry out experiments on MuLane. As shown in Table A8, when using ERFNet as the detection model, our method delivers 4.65% enhancement (87.93% vs. 83.28%) in contrast to the model using only the source domain data. Moreover, our method DACCA outperforms existing methods in accuracy, FP, and FN. Specifically, DACCA is 1.92% higher than PyCDA in accuracy (87.93% vs. 86.01%), 6.15% lower than Cross-domain in FP (25.95% vs. 32.10%), and 7.09% lower than PyCDA in FN (27.08% vs. 34.17%). All these results reflect the outperformance of our method.

