# OpenReview forum: "Context-Aware Unsupervised Domain Adaptive Lane Detection"
_ICLR.cc/2024/Conference — Submitted to ICLR 2024_

### Official Review · Reviewer_cVkk · 2023-10-23

**Soundness:** 3 good
**Presentation:** 3 good
**Contribution:** 2 fair
**Rating:** 5
**Confidence:** 4

**Summary:**

The paper presents an unsupervised domain adaptive approach for the lane detection task by utilizing a teacher-student framework and contrastive learning paradigm. Firstly, the method utilizes a positive sample memory module to retain domain-related lane features. Next, unlike existing methods, the authors propose cross-domain contrastive learning to improve feature discrimination. Additionally, a domain-level feature aggregation method is introduced to combine domain-level features with pixel-level features. The proposed unsupervised domain adaptive algorithm achieves the state-of-the-art performance on the TuLane, MuLane, and MoLane datasets.

**Strengths:**

1. Idea seems fundamentally sound, paper is well written.
2. Pixel-level feature aggregation module considers highly uncertain pixel of background class, which makes sense for me.
3. Experiementation results are provided to support the proposed method.

**Weaknesses:**

1. The experimentation settings in the paper are confusing as it is unclear which dataset is used as the source domain and which one is used as the target domain. This lack of clarity hinders proper understanding and evaluation of the proposed approach.
2. The paper fails to fully explore publicly available datasets from different domains. It lacks experimentation on datasets such as CuLane and OpenLane, which limits the generalizability and thoroughness of the findings.
3. The algorithm presented in the paper is only applicable to segmentation-based lane detection methods. This limitation reduces its potential contribution since most of today's algorithms tend to be either transformer-based or keypoint-based, making the proposed approach less relevant to current state-of-the-art techniques.

**Questions:**

While incorporating UBP into the feature aggregation module seems reasonable, the extent to which it contributes to the overall performance remains unclear.

---

> ### Author Response · Authors · 2023-11-22
> **Response to Reviewer cVkk**
>
> **Q1: The experimentation settings in the paper are confusing as it is unclear which dataset is used as the source domain and which one is used as the target domain. This lack of clarity hinders proper understanding and evaluation of the proposed approach.**
>
> REPLIES: We fully agree with the reviewers, and in response to this constructive comment, we have added a detailed description of the datasets used for the dataset source and target domains in Appendix A1.
>
> **Q2: The paper fails to fully explore publicly available datasets from different domains. It lacks experimentation on datasets such as CuLane and OpenLane, which limits the generalizability and thoroughness of the findings.**
>
> REPLIES: We totally agree with the reviewer. Following this constructive comment, we have supplemented the crossover experiments using different domain datasets to fully demonstrate the generalizability and performance of our method, including the transfer experiments between CULane and OpenLane. The experimental results are shown in Table 4 of Section 4.3 and Table A7 of Appendix A5. DACCA demonstrates superior performance compared to existing methods across all metrics, particularly when transferring from the CULane dataset to the Openlane dataset. It further confirms the generalization capability of our method.
>
> **Q3: The algorithm presented in the paper is only applicable to segmentation-based lane detection methods. This limitation reduces its potential contribution since most of today's algorithms tend to be either transformer-based or keypoint-based, making the proposed approach less relevant to current state-of-the-art techniques.**
>
> REPLIES: Here we would like to make further clarification. As pointed out in the review, in this work we mainly use a segmentation-based approach to illustrate the whole pipeline of the method, but our method has the potential to be a generalized domain adaptive framework for lane detection. Firstly, the anchor, positive sample, and negative sample in contrastive learning are not limited to pixels, but can be keypoints and queries in a transformer. Secondly, the DFA module is essentially a plug-and-play feature aggregation module. Our future work will validate the effectiveness and advantages of our approach by applying it to other lane line detection methods. We have pointed out it in the conclusion section.
>
> **Q4: While incorporating UBP into the feature aggregation module seems reasonable, the extent to which it contributes to the overall performance remains unclear.**
>
> REPLIES: We fully agree with the reviewer's comments. To highlight the contribution of UBP to the overall performance, we have supplemented the corresponding experiments in Section 4.2. The results given in Table 1 show that the feature aggregation of UBP can effectively improve Accuracy by 1.56%, reduce FP by 2.26% and FN by 2.79%. It also indicates that our method can effectively correct the features of UBP and thus improve the performance of the model.

---

### Official Review · Reviewer_1nVJ · 2023-10-31

**Soundness:** 3 good
**Presentation:** 3 good
**Contribution:** 2 fair
**Rating:** 3
**Confidence:** 4

**Summary:**

This paper proposes a Context-aware Unsupervised Domain-Adaptive Lane Detection (CUDALD) to improve feature discrimination and cross-domain knowledge transferring in the domain-adaptive lane detection field. To this end, the authors introduce two key components: cross-domain contrastive loss and domain-level feature aggregation. The former aims to effectively distinguish feature representations among categories while the latter combines domain-level and pixel-level features to enhance cross-domain context dependency. Extensive experiments on TuLane, MuLane and MoLane datasets verify the effectiveness of the proposed method.

**Strengths:**

- Aggregating the contextual information from the whole domain for pixel feature enhancement is logically sound and interesting. Despite its simplicity, this concept yields significant performance improvements compared to current state-of-the-art baselines.
- The authors conducted comprehensive comparison and ablation experiments on a wide range of benchmark datasets, confirming the effectiveness of their proposed approach.
- This paper is well-structured and well-explained. The method is accompanied by sufficient details and illustrative diagrams, such as Figures 1 and 2.
- Extensive visualization results and qualitative comparisons, e.g. Figures 4 and 5, further highlight the advantages of the proposed approach.

**Weaknesses:**

- Limited originality in cross-domain contrastive learning. This component heavily relies on the previous work (Wang et al., 2021; 2023) and provides an increment improvement (a positive sample selection). It doesn’t strike me as particularly novel.

- Combinatorial contribution. Each contribution addresses a specific issue in the unsupervised domain-adaptive field. The entire paper lacks content coherence.

- The title fails to effectively represent the contributions of the paper. 'Context' is a broad concept.
Here, the authors primarily propose aggregating domain-level context for pixel-level feature enhancement. Therefore, it is best to highlight this aspect.

**Questions:**

Please see the weakness.

---

> ### Author Response · Authors · 2023-11-22
> **Response to Reviewer 1nVJ**
>
> **Q1: Limited originality in cross-domain contrastive learning. This component heavily relies on the previous work (Wang et al., 2021; 2023) and provides an increment improvement (a positive sample selection). It doesn’t strike me as particularly novel.**
>
> REPLIES: We would like to further explain the originality of our work, and expect further kind and careful evaluation from the reviewer. The existing works (Wang et al., 2021; 2023) mainly focus on improving the form of contrastive loss, while the core of our proposed cross-domain contrastive loss is in the selection of samples without any changes to the contrastive loss. That is, we believe that the selection of samples in contrast learning is equally critical, so we re-select the anchor, positive sample, and negative sample involved in contrastive loss, thereby achieving better results.
>
> **Q2: Combinatorial contribution. Each contribution addresses a specific issue in the unsupervised domain-adaptive field. The entire paper lacks content coherence.**
>
> REPLIES: As the reviewer will note, two components, cross-domain contrastive loss and domain-level feature aggregation (DFA), of our approach are complementary and indispensable, not simply combined. The former is responsible for aligning features between different domains, and the latter focuses more on discriminative features inter-categories within a domain. The results of the ablation experiments show that the models are underperforming without either component, as shown in Figures A3 and A4. We sincerely hope the reviewers to give further consideration and re-evaluation to our approach.
>
> **Q3: The title fails to effectively represent the contributions of the paper. 'Context' is a broad concept. Here, the authors primarily propose aggregating domain-level context for pixel-level feature enhancement. Therefore, it is best to highlight this aspect.**
>
> REPLIES: We appreciate the keen insight of the reviewer, and have revised the title of the paper to “Unsupervised Domain Adaptive Lane Detection via Contextual Contrast and Aggregation”, which we believe can highlight the contributions of this work.

---

### Official Review · Reviewer_YuT7 · 2023-11-03

**Soundness:** 3 good
**Presentation:** 2 fair
**Contribution:** 2 fair
**Rating:** 3
**Confidence:** 4

**Summary:**

This paper proposes that ineffectively learning discriminative features and transferring knowledge across domains are two crucial issues in domain-adaptive lane detection task, and introduces a method named context-aware unsupervised domain-adaptive lane detection which is consisted of cross-domain contrastive loss and domain-level feature aggregation to tackle these issues.

**Strengths:**

1.	The incorporation of contrastive loss and feature aggregation strategies, as evidenced by the experimental results, appears to be a valuable addition to the paper.
2.	The experiment is sufficient.

**Weaknesses:**

1.	Lack of inspiration. While the article focuses on unsupervised domain-adaptive lane detection, it makes me confused whether its proposed issues, i.e., ineffectively learning discriminative features and transferring knowledge across domains could be applied to all unsupervised domain-adaptive segmentation tasks. It prompts consideration of whether the issues discussed are universal or specific to this particular domain. Additionally, the introduction of contrastive loss and feature aggregation, while useful, may not fully justify the article's contribution, as these methods are well-established in domain-adaptation field.
2.	The quality of writing and diagrams in the paper requires improvement. Several definitions provided in the article lack clarity, hindering reader comprehension. For instance, in section 3.2, the term 'anchor' is introduced without a precise definition. Additionally, the role of modules is not adequately explained, such as how the introduction of PSMM ensures the appropriate assignment of positive samples. Furthermore, the figures lack detail and refinement, and the overall layout, as seen in Figure 2, does not appear to be carefully designed.
3.	Some conclusions of the article are exaggerated. For example, in section 4.3, the authors claim that their method performs significantly better than other methods when using ERFNet as the detection model, but choosing PyCDA (Accuracy/%: 86.73) and MLDA(Accuracy/%: 88.43) to compare instead of SGPCS with the accuracy of 89.28%.

**Questions:**

See the Weaknesses.

---

> ### Author Response · Authors · 2023-11-22
> **Response to Reviewer YuT7**
>
> **Q1: Lack of inspiration. While the article focuses on unsupervised domain-adaptive lane detection, it makes me confused whether its proposed issues, i.e., ineffectively learning discriminative features and transferring knowledge across domains could be applied to all unsupervised domain-adaptive segmentation tasks. It prompts consideration of whether the issues discussed are universal or specific to this particular domain. Additionally, the introduction of contrastive loss and feature aggregation, while useful, may not fully justify the article's contribution, as these methods are well-established in the domain-adaptation field.**
>
> REPLIES: We appreciate the reviewer’s expertise and insight. As the reviewer will observe, upon our redistilling, the contribution and inspiration of this paper lies in the selection of samples in contrast learning and the aggregation of effective features. It helps to improve feature discriminative and efficient knowledge transfer across domains in unsupervised semantic segmentation domain adaptive tasks. First, ineffectively learning discriminative features and transferring knowledge across domains are significant challenges encountered in all unsupervised semantic segmentation domain adaptation tasks, including lane detection. Second, although cross-domain contrastive loss has been widely used in unsupervised semantic segmentation domain adaptation tasks, the existing methods focus on the loss function. In contrast, we propose a novel sampling strategy to fully utilize the potential of contrastive loss without modifying the existing contrastive loss. Third, for feature aggregation, to our best knowledge, all existing methods currently perform feature aggregation within a mini-batch. We take a new perspective, i.e., feature fusion using domain-level features, instead of feature aggregation on a mini-batch or an image alone. As Figure A3 in the Appendix demonstrates, domain-level features can better align features between different domains.
>
> They are the main contributions of our approach, which we believe will give, to some extent, inspiration to the study of sample selection in contrastive learning and how to select effective features for feature aggregation. We sincerely hope the reviewer to re-evaluate it.
>
> **Q2: The quality of writing and diagrams in the paper requires improvement. Several definitions provided in the article lack clarity, hindering reader comprehension. For instance, in section 3.2, the term 'anchor' is introduced without a precise definition. Additionally, the role of modules is not adequately explained, such as how the introduction of PSMM ensures the appropriate assignment of positive samples. Furthermore, the figures lack detail and refinement, and the overall layout, as seen in Figure 2, does not appear to be carefully designed.**
>
> REPLIES: As the reviewer will note, we have done our best to improve the quality of the writing and images in the revision, including pointing out that the anchor is the pixel feature being compared, i.e., the candidate pixel, the feature representation in PSMM is more accurate than pixel-level feature representation, providing more positive signals in the contrastive loss, as well as the improvement of the appearance of each module in the Student-Teacher model structure, as shown in Figure 2.
>
> **Some conclusions of the article are exaggerated. For example, in section 4.3, the authors claim that their method performs significantly better than other methods when using ERFNet as the detection model, but choosing PyCDA (Accuracy/%: 86.73) and MLDA(Accuracy/%: 88.43) to compare instead of SGPCS with the accuracy of 89.28%.**
>
> REPLIES: We totally agree with the reviewer. We have revised and clarified the inaccurate expressions in this manuscript.

---

### Official Review · Reviewer_ZmgM · 2023-11-05

**Soundness:** 3 good
**Presentation:** 3 good
**Contribution:** 3 good
**Rating:** 6
**Confidence:** 4

**Summary:**

In this work, author target the domain adaption in lane line detection, identifying two specific problems in prior work and proposes corresponding solutions. Firstly, prior work only focus on pixel level feature and ignore inter-lane feature presentation. Secondly, prior work don’t adopt cross domain representation for adoption. Authors propose Context-aware Unsupervised Domain-Adaptive Lane Detection (CUDALD) framework, including a contrastive learning function by redesign the sampling method, and a pixel-domain feature aggregation model to enrich representation. Authors shows that the proposed method achieved SOTA on multiple dataset.

**Strengths:**

1. Author have targeted a very specific, often ignored problem, domain adaption for lane line detection, which is a critical problem in modern ADAS.
2. Author identify two problems in prior work adopting solution from a general domain(eg, domain adaption for segmentation), and propose solution specific designed for laneline domain adaption, showing improvement over prior art.

**Weaknesses:**

1. There’s limited discussion, comparison and analysis over prior work that also target specifically on laneline domain adaption, for instance, MLDA(Li et al., 2022). The statement in introduction that 'Unfortunately, achieving cross-domain context dependency in domain-adaptive lane detection remains unexplored.' is also less accurate as prior works like MLDA already explored this domain.
2. Author only conduct experiment on TuLane, MuLane, and MoLane, three simulated lane datasets proposed from one single work, the result would be more convincing if authors could conduct domain transfer experiments between Tusimple and Culane, which are more commonly used dataset for the community.
3. This work is a bit hard to follow, for instance, in 3.3, author use ‘features from a whole domain is more beneficial’ as a core argument over prior work, without clearly pointing to ‘what is whole domain and how does the method align with whole domain’ in the following implementation. The implementation is also hard to follow without clearly define ‘lane feature and domain feature’ in the first place.
4. The conclusion of ‘with more advanced lane detection methods, e.g., anchor-based methods’ is also lack of context. Firstly, in the related work, author stated that ‘In this paper, we consider segmentation-based domain-adaptive lane detection.’, which contradict with the conclusion. Secondly, author hasn’t provide why context why ‘author based’ method is superiority over other method’.
5. Author mentioned the term PSMMs for multiple time in the introduction and related work, without explicitly explain its functionality. It would be easier to follow if author could briefly explain its functionality earlier in the paper.
6. Please also include proper citation of work for table4/5, and the last two page of the paper.

**Questions:**

it's still less clear to me what the motivation and implementation of cross-domain contrastive loss. It seems like the only change author proposed is a novel sampling method, please make this clear.

---

> ### Author Response · Authors · 2023-11-22
> **Response to Reviewer ZmgM**
>
> **Q1: There’s limited discussion, comparison and analysis over prior work that also target specifically on laneline domain adaption, for instance, MLDA(Li et al., 2022). The statement in introduction that 'Unfortunately, achieving cross-domain context dependency in domain-adaptive lane detection remains unexplored.' is also less accurate as prior works like MLDA already explored this domain.**
>
> REPLIES: Following the reviewer's comment, we have added a review of methods such as MLDA in the introduction section and supplemented a comparison and discussion with MLDA in the experimental section (Sect. 4.3). At the same time, we have revised and clarified the inaccuracies in Introduction.
>
> **Q2: Author only conduct experiment on TuLane, MuLane, and MoLane, three simulated lane datasets proposed from one single work, the result would be more convincing if authors could conduct domain transfer experiments between Tusimple and Culane, which are more commonly used dataset for the community.**
>
> REPLIES: Following this constructive comment. In this revision, we have supplemented the domain transfer experiments between CULane and Tusimple, detailed in Appendix A.5.
>
> **Q3: This work is a bit hard to follow, for instance, in 3.3, author use ‘features from a whole domain is more beneficial’ as a core argument over prior work, without clearly pointing to ‘what is whole domain and how does the method align with whole domain’ in the following implementation. The implementation is also hard to follow without clearly define ‘lane feature and domain feature’ in the first place.**
>
> REPLIES: As the reviewer will note, we have made our best to clarify and explain some confusing concepts and terms. Specifically, we carefully define lane features and domain features in this revision. The former refers to pixel-level features of a lane, while the latter refers to all lane-line pixel features in the source and target domains. In addition, whole domain features refer to the pixel-level features of lanes in all images in a domain. Since DFA directly fuses domain features into individual lane line pixel features, it enables the model to learn domain-level features better and thus better align with the whole domain.
>
> **Q4: The conclusion of ‘with more advanced lane detection methods, e.g., anchor-based methods’ is also lack of context. Firstly, in the related work, author stated that ‘In this paper, we consider segmentation-based domain-adaptive lane detection.’, which contradict with the conclusion. Secondly, author hasn’t provide why context why ‘author based’ method is superiority over other method’.**
>
> REPLIES: As pointed out in this review comment, the expression in our conclusions was inappropriate and has been revised. We plan to apply our framework to other types of lane detection methods, e.g., anchor-based, and transformer-based approaches, to verify its generality in the future.
>
> **Q5: Author mentioned the term PSMMs for multiple time in the introduction and related work, without explicitly explain its functionality. It would be easier to follow if author could briefly explain its functionality earlier in the paper.**
>
> REPLIES: Following this constructive suggestion, we have added to Introduction a description of the functionality of PSMMs, i.e., saving domain features of each line.
>
> **Q6: Please also include proper citation of work for table4/5, and the last two page of the paper.**
>
> REPLIES: We have cited proper work for Tables 4, A8, and A9 in the revision.
>
> **Q7: it's still less clear to me what the motivation and implementation of cross-domain contrastive loss. It seems like the only change author proposed is a novel sampling method, please make this clear.**
>
> REPLIES: The motivation for proposing cross-domain contrastive loss is to achieve high effectiveness in learning discriminative features. As demonstrated in Figure A4, cross-domain contrastive loss has solved the problem that cross-domain feature dependency only aligns inter-domain features but does not discriminate well inter-category features without modifying the loss function. We have proposed a new sampling policy to implement the cross-domain contrastive loss. In contrastive learning, sample selection is very important, while existing works in (Wang et al., 2021, 2023), only consider positive sample selection but disregard the selection of anchor and negative samples.

---

> > ### Comment · Reviewer_ZmgM · 2023-11-22
> > **Update Soundness to 3 and my score to 6.**
> >
> > In general I am satisfied author's reply for addressing many concerns, and see how author's revision make the work toward a good direction.
> > As some other reviewer also pointed out, the experiment settings is confusing, especially for people not work in this field. I suggest author to include a section of 'related work' on prior work working on 'domain adaption of lane line'( (Garnett et al., 2020), (Gebele et al., 2022) and MLDA) to provide some background for this domain. The focus of this work should be 'adopting domain adaption to lane detection', rather than 'proposing a novel general domain adaption method'. Thus I feel like author should focus more on explaining the field you're targeting(aka, domain adaption on lane detection, and what is the problem of applying general domain adaption to this field, and what problem does prior work exist(like MLDA only use single frame information as mentioned in draft, etc)).
> >
> > This work will be in much a better shape after author reflecting promised changed in point 4, and revised clarity for point 4 and 7. Please also consider moving experiment of Tusimple to main paper as it reflect experiment on one of the most widely used dataset.
> >
> > I will update Soundness to 3 and my score to 6.

---

> > > ### Author Response · Authors · 2023-11-23
> > > **Further response to Reviewer ZmgM**
> > >
> > > Thank you for updating your score and the soundness. We give further response to your latest comments.
> > >
> > > **Q1: Include a section of 'related work' on prior work working on 'domain adaption of lane line'( (Garnett et al., 2020), (Gebele et al., 2022) and MLDA) to provide some background for this domain and explaining the field you're targeting.**
> > >
> > > REPLIES: We fully agree with the reviewer and have added a subsection, Unsupervised domain adaptive lane detection, to Related work.
> > >
> > > **Q2: Please also consider moving experiment of Tusimple to main paper as it reflect experiment on one of the most widely used dataset.**
> > >
> > > REPLIES: Following this constructive comment, we have moved the experiment of Tusimple to the main paper.

---

### Author Response · Authors · 2023-11-22

We would like to express our sincere thanks to the reviewers for their insightful comments and helpful suggestions for the improvement of our paper. We have made a thorough modification, addressing the concerns of the reviewers one by one in this revision. Please note that we have highlighted the newly added places in blue and modified places in red in the revision.

---

### Meta-Review · Area_Chair_FheQ · 2023-12-23

**Metareview:**

This paper presents a method for domain adaptation for lane detection. The paper proposes a method containing 2 components, a "cross-domain contrastive loss" and "domain-level feature aggregation". The reviewers note that the experimentation show the advantage of the approach. However, there were several concerns on the paper raised by YuT7, 1nVJ, cVkK, including quality and clarity of writing (YuT7, cVkK), a limited contribution to a problem of narrow scope (YuT7, 1nVJ, cVkK), and overall concerns regarding the experimentation (YuT7, cVkK). The initial reviews were majority reject (3 reject, 1 accept). While the authors submitted a thoughtful response, the majority of reviewers did not revise their responses. The AC agrees with the majority decision that the paper's contribution is currently not above the bar for ICLR.

**Justification For Why Not Higher Score:**

The AC agrees with the majority decision that the paper's contribution is currently not above the bar for ICLR.

**Justification For Why Not Lower Score:**

N/A

---

### Decision · Program_Chairs · 2024-01-16

Reject